# Happiness and Mental Disorders

**DOI:** 10.3390/healthcare10091781

**Published:** 2022-09-16

**Authors:** Eva Lourdes Díaz Hernández, Pedro Ruymán Brito Brito, Alfonso Miguel García Hernández

**Affiliations:** 1Community Mental Health Unit of San Cristóbal de la Laguna, Canary Islands Health Service, 38204 San Cristóbal de La Laguna, Spain; 2Primary Care Management of Tenerife, Canary Islands Health Service, University of La Laguna, 38010 San Cristóbal de La Laguna, Spain; 3Nursing Department, Faculty of Health Sciences, University of La Laguna, 38010 San Cristóbal de La Laguna, Spain

**Keywords:** happiness, mental disorder, mental health, nursing, quality of life

## Abstract

The research object is the approach the meaning of happiness for people with severe mental illness (SMI) under follow-up in a mental health unit in Tenerife. The research aims to improve the care they receive. This qualitative, phenomenological study uses convenience and intentional sampling. Questions were administered to a focus group consisting of 4 women and 1 man, aged 35–69, and 16 individual interviews were conducted with 8 women and 8 men, aged 20–62. The interviews were audio-recorded, with prior consent, transcribed verbatim, coded, and analyzed using QSR N-Vivo Release 1.4.1 (851), Spain. Happiness has three dimensions: personal, interpersonal–relational, and temporal. The personal dimension includes personality, positive emotions, health, motivations for establishing personal goals, and engaging in activities. The interpersonal–relational dimension includes family support; social support and relationships; social and occupational functioning; overcoming deaths, breakups, or job losses; and the absence of stigma on mental illness. The temporal dimension establishes that happiness can be comprised of either a set of happy moments or a continuous state of happiness that varies throughout life. Based on the results of this research, it could be proposed that future research should focus on the effectiveness of nursing interventions, addressing the life goals of people with mental disorders, and the pursuit of their happiness.

## 1. Introduction

In the World Health Organization’s [1] definition of “health,” “mental health” is included as “a state of complete physical, mental and social well-being, and not merely the absence of disease or infirmity”.

In the Canary Islands Mental Health Plan 2019–2023, the Canary Health Service [2] proposes, in relation to the care of people with severe mental illness (SMI), different strategies to maintain and ensure the continuity of care through the service network, reduce the number of hospitalizations, and improve social functioning and quality of life. Furthermore, the December 14 Law 39/2016, titled “Promotion of Personal Autonomy and Attention to People in a Situation of Dependency” [3], contemplates aspects to favor or promote integration and reduce stigmatization. The European Action Plan for Mental Health 2013–2020 includes the global aim of promoting mental well-being; preventing mental disorders; providing care; improving recovery; promoting human rights; and reducing mortality, morbidity, and disability among people with mental disorders. This plan takes a comprehensive, multisectoral approach, coordinates the health and social sector services, and proposes key indicators and goals that are useful for evaluating implementation, progress, and impact. Similarly, the action plan is based on the globally accepted principle that “There is no health without mental health” [1].

Historically, the approach to the study of happiness spans from ancient times to the consolidation of psychology as a science toward the end of the 19th century. Given the various scientific approaches from fields such as philosophy, theology, sociology, and the social sciences, it is difficult to singularly define “happiness” [4]. However, there are few published studies focusing on SMI and happiness although there has been increasing empirical attention paid to broad positive constructs, such as quality of life, life satisfaction, and subjective well-being [5,6].

Seabra et al. [7] concluded that their study participants with mental illness did not express in-depth experiences of happiness according to the definitions. However, the factors that contribute to increasing happiness are personal, familial, social, and emotional support, while those that contribute to reducing it are personal factors, medication side effects, lack of social support, and affective and emotional deficits. Crucial aspects for understanding the perception of happiness in people with mental illnesses include interventions that promote rehabilitation, social relationships, and the meaning of life. These allow for greater perceived happiness, which is essential for recovery [7].

In the personal dimension, the factors that contribute to diminishing happiness are medication, which is positively valued, although it is associated with various problems (e.g., weight control, sexual dysfunction, etc.); taking medication, which is still associated with the fear of not being considered “normal” [8]; mental disorders or dual pathologies [9,10,11]; the chronicity of the mental disorder [10]; the perception of increased stress [12]; and depression and negative symptoms [13]. In the interpersonal dimension, a surge in factors related to social support, fear, loneliness, and isolation [8,14], and the absence of occupational activities and social dysfunction [14] have been observed. Decreased happiness is more related to aspects of the personal domain and less related to interpersonal ones. Some authors have argued that the perception of happiness is mainly personal; that is, it is more related to the way a person experiences specific external events and less linked to the events themselves [12,15]. Lack of support is not considered to be a factor that promotes unhappiness [10] although symptoms and mental pathology are determining factors for a lesser perception of happiness [7,8].

Schizophrenia negatively affects happiness [16], but happiness can still be an achievable goal for people with this disorder [17]. This finding confirms Saks’s [18] claim that people with schizophrenia can find “well-being within the disease”.

There are few studies specifically focused on understanding happiness in severe mental disorders [8,12,13,16,17,19]. Some studies have found lower levels of happiness in patients diagnosed with schizophrenia [7,8,12,16,17]. However, other studies report that people with mental disorders have levels of happiness comparable to those of people without mental disorders [10,13,20]. Similarly, no differences were found according to sociodemographic variables, duration of illness, severity of positive and negative symptoms, physical status, comorbidities, or cognitive functioning [7,19]. However, happiness increases social, emotional, and moral capacity. People with mental illness who are predisposed to good mental health have a positive perception of happiness and feel happier [7,19]. Furthermore, Bergsma et al. [5] in their review concluded that happiness can be trained and therefore increased.

It is difficult to investigate the meaning of happiness using quantifiable measures and instruments; however, the analysis of narratives, from a constructivist and phenomenological perspective, provides relevant information that could help us to achieve our goal of getting closer to the meaning of happiness.

Happiness is important in achieving recovery in severe mental disorders [7]; therefore, this study aims to approach the meaning of happiness for these people in order to improve their health as their responses can guide nursing interventions.

## 2. Materials and Methods

In this study, qualitative research was carried out, based on a constructivist and interpretive phenomenological analytical perspective, through convenience and purposive sampling involving a focus group and individual interviews. The interview questions were designed to reflect the research objective, that is, to understand what happiness means to people with SMI (see Table 1).

The interviewees were persons over 18 years of age who had follow-up nurse consultations in a mental health unit in Tenerife. For participant selection, convenience and purposive sampling were employed, taking into account the following inclusion criteria: having an SMI diagnosis, not having an intellectual disability, not presenting with psychopathological decompensation, and not being legally incompetent. Participation was voluntary, and all participants were guaranteed confidentiality and anonymity. Participants’ permission was sought prior to audio-recording the interviews, and all participants signed an informed consent form, which provided greater detail about the study. In addition, age, sex, and type of mental disorder were taken into account (see Table 2).

Complementary methods were used, with similar epistemological approaches, in which data were collected through different methods: a focus group, individual interviews, and observations collected in the field diary, providing parallel data and allowing for triangulation (see Figure 1).

The focus group interview had a duration of 90 min and involved 5 participants (from P0.1 to P0.5; see Table 2), with the main researcher acting as the interviewer/moderator (research-qualified and experienced in clinical interviewing) and a mental health nurse specialist as an observer. The interview took place in a cafeteria outside the health center. This location was chosen because it was a neutral, comfortable, low-noise space with no interference from other clients or staff.

The individual interviews (from P1.1 to P1.15 and P0.0; see Table 2) were carried out in the nursing consultation room at the mental health unit. This space created a climate of trust and collaboration, thanks to the previously established therapeutic relationship. The individual interviews ranged from 20 to 50 min in duration. They were audio-recorded and transcribed verbatim for subsequent coding and analysis.

The transcribed text was fragmented, and different codes were established through an inductive process that involved constant comparison. Groups were formed according to categories representing the broader concepts associated with each code. From individual interview 15, no new codes were identified based on the participants’ responses. N-Vivo Release 1.4.1 was used to manage the interviews, establish the codes, and group them into categories and subcategories. The printed interviews and N-Vivo tools were used for the coding and interpretive phase of the analysis. The results were triangulated with other professionals.

## 3. Results

Based on interviews with 21 participants, 3 dimensions were established: the interpersonal–relational dimension, the personal dimension, and the temporal dimension (see Table 3). Subcategories were also established within each dimension. Taken together, these reflect the meaning of happiness for the interviewees with SMI.

The hierarchical map indicates the importance of the various categories and subcategories, according to the frequency of the coded responses (see Figure 2).

In-depth analysis of each dimension showed that, in the personality subcategory within the personal dimension, personal qualities, traits, strengths, and values have a positive character, characterized by optimism, independence, solidarity, and acceptance, and some are even fundamental to achieving self-realization, creativity, satisfaction with life, and spirituality–religiosity.

Optimism: “Well yes, I still have the ability to be surprised by life. I think that should not be lost” (P1.1). “That’s my vision and attitude and that’s what you have to work on. You have to believe in yourself”.(P1.12)

Independence: “I think you are really happy when you detach yourself from all that, including your family. […] I no longer expect anything from anyone. And not expecting anything from anyone allows me to be myself”.(P1.1)

Solidarity: “Joy, joy and I try to pass it on, because I know what has happened and I don’t want someone else to go through it”.(P0.2)

Self-realization: “I try to do what I like without a productivity vision; dancing, singing, and rhythmic gymnastics mixed with ballet, this is what I get”.(P1.15)

Acceptance: “Sometimes I have a goal and things do not go my way, now I have learned to accept things more and wait. Everything happens for a reason” (P1.12). “I have recovered, and I have lost. You do not always win; you also lose”.(P1.8)

Creativity: “I try to do what I like and without a vision of productivity, dancing, singing, rhythmic gymnastics mixed with ballet, that’s what I get out of it”.(P1.15)

Satisfaction with life: “So that your day-to-day life is somehow pleasant, entertaining, fun. That you feel good in the routine you are in”.(P1.5)

Personal knowledge and the identification of emotions are necessary to reduce pain and suffering and to achieve happiness. People’s emotions and how they handle them are strongly related to their perception of happiness, especially for negative emotions that cause unhappiness. As shown in Figure 3 and Figure 4, positive emotions remarkably occupy a large part of people’s memories. This is especially true for joy, with which everyone is familiar, as well as for peace, tranquility, love, freedom, and fulfillment (see Figure 3). On the other hand, sadness and pain are the main negative emotions associated with suffering; these are accompanied by loneliness, helplessness, pessimism, worry, guilt, and confusion (see Figure 4).

Health is an important subcategory in the personal dimension. It is understood that when a person is in poor health, recovery is essential. However, with SMI, the importance of the symptoms, the functional deterioration produced, and the side effects of the medication, which often make it difficult to forget or disidentify with the disease, as is the self-stigma from which some persons, including some of the present study’s participants, suffer. The disease’s presence is pronounced in some cases, while it is not so obvious in others.

Disease: “[For] 10% of the day, I am crazy, and 90% of the day, I am doing things”.(P1.15)

Treatment: “Man, what I am doing now: go to the psychologist, follow my psychiatric therapies; I am very compliant. All of this has given me stability”. (P1.4)

Recovery: “I think that I have returned to how I was before…”.(P1.5)

Stigma: “That they better understand the disease, without stigmatizing it so much”.(P1.4)

The motivation subcategory includes actions aimed at achieving happiness. Despite the difficulties, deficits, or scarcity of resources that may be present in participants’ lives, personal growth is fundamental, as are establishing vital goals (i.e., having a job, a partner, children, traveling, being happy, etc.), and engaging in fulfilling activities. Regarding the latter, these do not have to be big activities as that is sometimes not possible, but one should try to enjoy the little everyday things, without forgetting that none of this can be achieved if the basic needs are not covered, perhaps with the addition of something whimsical that increases the number of happy moments.

Personal growth: “It has made me discover myself internally, which I might not have done otherwise”.(P0.4)

Material happiness: “Happiness is in the small things, not in having millions” (P1.11). “Being able to do normal things. Because I would like to do things that I cannot do for lack of money. I would like to study abroad; I would like to do a lot of things. But I cannot––that is why I aspire to the ordinary, the common”.(P1.5)

Basic needs: “Having a roof over my head to protect me, food to feed me and clothes so I don’t go out naked in the street, that’s basically it”.(P.1.1)

In the interpersonal–relational dimension, relationships with others, family and social support, and optimal social and occupational functioning all play an important role. The participants made it very clear that without stigma and losses, they would definitely be happier.

Family support: “My family was always there… Unconditional[ly]”.(P1.2)

Social support: “The neighbors have turned to me. We had problems in the community, but when that happened, and my parents passed away … We do not have many resources and decided to start from scratch, and do you know what? They worry about me; when I arrive, when I leave, we text each other through WhatsApp. My neighbors are my family, and on Saturdays, I make coffee and tell my neighbor. I buy some chocolates and I bring out some wine, and we take some pictures”.(P0.2)

Social and occupational functioning: “Keep the mind occupied, stimulate the mind […] and also the opposite: have time for rest and leisure”.(P1.5)

Stigma: “No, when they assign you a tag, they take things from you […] It is not the same when they treat you without knowing whether you have a disease or not”.(P1.1)

The third dimension, the temporal dimension, establishes that happiness can be perceived either as a set of happy moments or as a continuous state of happiness that varies throughout life.

Moments: “These are moments that we have in life […]. That is why I say [you should] have the greatest number of happy moments possible”.(P0.4)

Continuum: “[…] For me it is feeling good and bad; that is, I do not consider that I always feel good or always feel bad. Every day is like good and bad […] I do not consider it neutral, because I never finish conceiving happiness. Today, I am good and bad; there are things that are good and others that are bad, and there are days that are worse or better”.(P1.15)

## 4. Discussion

The study participants’ responses indicate that the meaning of happiness for people with SMI varies widely. The answers are not very homogenous, as demonstrated by the many subcategories the coding yielded and the establishment of two large categories representing the personal and interpersonal–relational dimensions, in line with the results of the review by Seabra et al. [7]. A third dimension was also identified, namely the temporal dimension, which is related to the perception of happiness in people’s lives from a temporal point of view. This clearly distinguishes the people who experience happiness as moments from those who understand it as a continuum that varies throughout life.

Regarding the personal dimension, it can be said that happiness depends on the qualities and values that define the person [12,15]. These belong to the personality subcategory, which includes optimism, independence, creativity, acceptance, and solidarity. This category also includes the need to be in an optimal state of health [8,9,10,11,12,13,16]. To be happy, people also need to be motivated to meet their basic needs, engage in pleasant activities, and set vital objectives. It should be noted that vital goals and motivations could change with the onset of SMI [5]; however, as Agid et al. [13] have pointed out, this is not always taken into account in recovery strategies. In contrast, the interpersonal–relational dimension includes relational factors, such as family and social support, optimal social and occupational functioning, and relationships, which combat the stigma and the losses suffered. This confirms that people with the same illness (schizophrenia) do not necessarily perceive happiness in the same way [8]. Furthermore, Palmer et al. [12] reported considerable heterogeneity between the happiness levels of participants with chronic schizophrenia. In their systematic review, Seabra et al. [7] found differences in the perception of happiness in people with (versus those without) a mental disorder, with the results fluctuating between 2% and 27%; furthermore, Gutiérrez-Rojas et al. [16] concluded after comparing people with schizophrenia with healthy people that the former showed lower levels of subjective happiness, well-being, and life satisfaction.

In accordance with the results of this research, some authors have concluded that certain characteristics define a happy person. The present study has demonstrated the importance of a positive, optimistic personality, as well as the ability to move on, avoid dwelling on negative things, count on social support, live in a society with a certain degree of economic development, and have the personal resources to achieve vital objectives, in addition to being satisfied in other areas, such as work and family [7].

Seabra et al. [7] did not find a single definition of happiness. They therefore concluded that happiness cannot be understood separately from a person’s social experience, well-being, recovery, and internal protective factors. They established that the factors that favor the perception of happiness can be encompassed in two dimensions: the personal dimension, which is more subjective and related to the self, and the interpersonal dimension, which is developed in multiple contexts (i.e., family, social, and emotional support). The results of this research clearly identified these two dimensions in the codings, in addition to the time dimension and the various subcategories that comprise each dimension (see Table 3). After coding, in the personal dimension, as shown on the hierarchical map (see Figure 2), the importance of personality and its relationship with happiness was highlighted, as well as aspects related to the motivation subcategory, which includes vital goals, personal growth and self-realization, health, and emotions [5]. This is consistent with Ferraz et al. [15], who concluded that happiness depends more on psychic aspects and a person’s attitude toward life than on external factors. Similarly, some of the codes obtained in this research agree with those of Palmer et al. [12], highlighting optimism, recovery, and independence, in addition to the importance of social and family support, as Greenberg et al. [21] described. Moreover, the importance of positive emotions, commitment, relationships, meaning, and achievements is emphasized in the research on happiness in the 21st century [22,23].

Salas and Garzón [24] have argued that, in order to achieve happiness and personal fulfillment, one must have a certain quality of life and one’s basic needs must be met. In the present research, only a few participants reported facing difficulties in these areas. However, the basic needs and material happiness subcategories confirm Salas and Garzón’s [24] results, also allowing us to recall Abraham Maslow’s [25] theory of human motivation.

Among the limitations were difficulties in relating our results with those obtained by other studies due to the heterogeneity of our participants and the scarcity of specific research on people with severe mental disorders and happiness, finding mostly research on people with schizophrenia or psychosis. Furthermore, the impossibility of finding a single definition of happiness reflected in the research with the use of other concepts, such as well-being, quality of life, life satisfaction … has made it difficult to find previous studies focused on happiness and severe mental disorders although that has enriched the results obtained in this research.

## 5. Conclusions

Regarding the results of our study, happiness includes three dimensions: the personal dimension, the interpersonal–relational dimension, and the temporal dimension. The personal dimension comprises personality, positive emotions, health, the motivation to set personal goals, engaging in activities, and having one’s basic and material needs covered so as to be able to enjoy happy moments. In the interpersonal–relational dimension, happiness depends on family and social support, relationships, social and occupational functioning, the overcoming and/or acceptance of losses such as deaths, breakups, or job losses, and the absence of stigma on mental illness. Finally, the temporal dimension establishes that happiness can be formed either by a set of happy moments or by a continuous state of happiness that varies throughout life.

This approach to defining happiness should guide objectives, strategies, and nursing interventions toward recovery and the pursuit of happiness, with a focus on personal motivation and goals.

## Figures and Tables

**Figure 1 healthcare-10-01781-f001:**
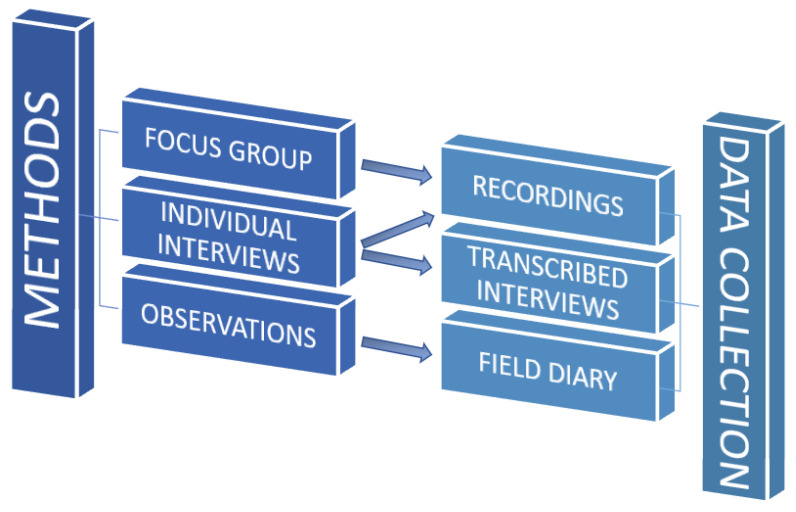
Methods and data collection.

**Figure 2 healthcare-10-01781-f002:**
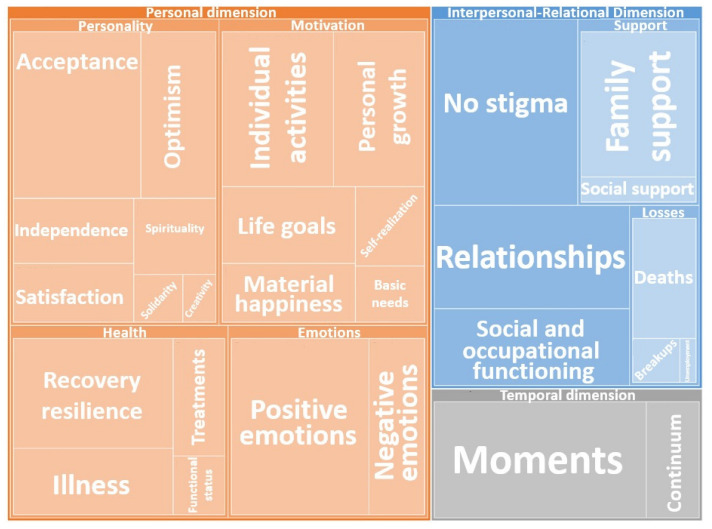
Concept: hierarchical map.

**Figure 3 healthcare-10-01781-f003:**
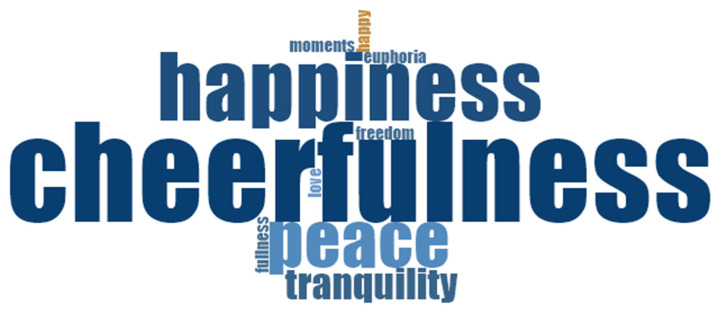
Positive emotions related to happiness: word cloud.

**Figure 4 healthcare-10-01781-f004:**
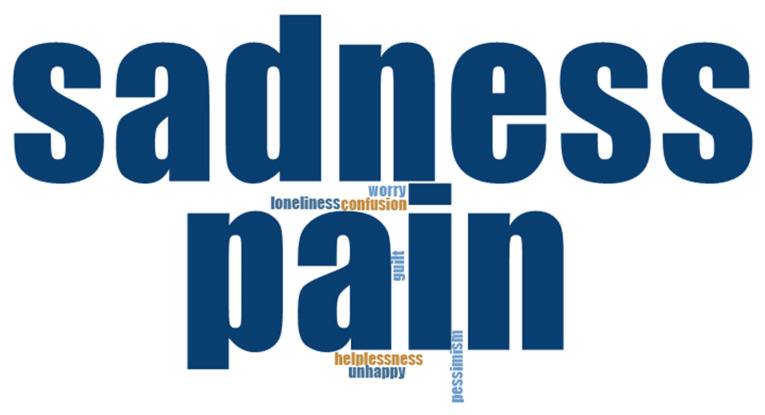
Negative emotions related to unhappiness: word cloud.

**Table 1 healthcare-10-01781-t001:** Interview.

1. What is happiness?
2. Are you happy or unhappy?
3. What would make you happy?
4. What are the components of happiness?
5. Do you remember a happy moment? What did you feel?
6. Do you remember an unhappy moment? What did you feel?
7. What changes/situations/moments made you go from happy to unhappy?
8. What do you do to be happy?
9. What should the world be like for you to be happy or happier?
10. What should you be like/should you change to be happy or happier?
11. Is anyone happy irrespective of the illness they have?
12. Do you think people are just as happy if they have a mental disorder?
13. What did the illness mean to you? Did it change your life? Does it have a meaning?
14. What factors changed?
15. Do you feel rejected?
16. How do you shield yourself from rejection?
17. Do you think that going to mental health centers is discriminatory?

**Table 2 healthcare-10-01781-t002:** Profile of participants.

Identification	Sex	Age	Severe Mental Disorder
(P0.0)	Male	47	Schizophrenia
(P0.1)	Female	69	Depressive disorder
(P0.2)	Female	62	Delusional ideas disorder
(P0.3)	Female	35	Delusional ideas disorder
(P0.4)	Male	42	Borderline personality disorder
(P0.5)	Female	52	Borderline personality disorder
(P1.1)	Female	36	Borderline personality disorder
(P1.2)	Male	43	Bipolar disorder
(P1.3)	Male	58	Paranoid schizophrenia
(P1.4)	Female	40	Bipolar disorder
(P1.5)	Male	20	Psychotic episode
(P1.6)	Male	56	Bipolar disorder
(P1.7)	Female	57	Schizophrenia
(P1.8)	Male	56	Paranoid schizophrenia
(P1.9)	Male	62	Personality disorder
(P1.10)	Male	44	Schizophrenia
(P1.11)	Female	60	Delusional ideas disorder
(P1.12)	Female	37	Schizophrenia
(P1.13)	Female	52	Paranoid schizophrenia
(P1.14)	Female	26	Schizophrenia
(P1.15)	Female	22	Borderline personality disorder

**Table 3 healthcare-10-01781-t003:** Concept of happiness: categories and subcategories.

Categories		Subcategories	
**CONCEPT**	INTERPERSONAL-RELATIONAL DIMENSIONS	No stigma	
		Social and occupational functioning	
		Relationships	
		Support	Family support
			Social support
		Losses	Deaths
			Unemployment
			Breakups
	PERSONAL DIMENSION	Motivation	Individual activities
			Self-realization
			Personal growth
			Material happiness
			Basic needs
			Life goals
		Emotions	Negative emotions
			Positive emotions
		Personality	Acceptance
			Creativity
			Spirituality
			Independence
			Optimism
			Satisfaction
			Solidarity
		Health	Illness
			Recovery-Resilience
			Treatments
			Functional status (Cognitive and physical)
	TEMPORAL DIMENSION	Continuum	
		Moments	

## Data Availability

The data presented in this study are available upon request from the corresponding author.

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
