# Peer review of "Happiness and Mental Disorders"

_healthcare, 2022, doi:10.3390/healthcare10091781_

Round 1
Reviewer 1 Report
Dear Authors,
Congratulations your research! It's very important topic especially in the context of the global aging population.
I have only 2 comments to be included:
Comments:
1) Please include in the methodology part (Materials and methods)
Where was the research conducted? Please describe the place: Tenerife or several Canary Islands (which one)
2) The limitations of research are not described, Please include in the end of the text.
Overall Recommendation
I recommend this paper to publish after minor correction
Author Response
First of all, I would like to thank you for your suggestions for improving the article. In response to each of these suggestions, the initial document has been modified, and I attach the document for further review. Thank you in advance, I look forward to your new assessment soon.
Kind regards

Reviewer 2 Report
Abstract
1. A sentence should be written at the end of the text that will shed light on the future studies.
Introduction
2. A reference should be added to the following sentence.
“Historically, the approach to the study of happiness spans from ancient times to the consolidation of psychology as a science toward the end of the 19th century”.
3. There are paragraphs of 2 sentences. These paragraphs should be combined in a flow.
4. At least 2 references should be added to sentences like the one below.
“Many studies have found that the difference between the perception of happiness in people with versus without mental disorder is small and statistically insignificant, and that the necessary conditions for happiness are similar”.
“In some studies, no differences were found according to sociodemographic variables, duration of the illness, severity of the positive and negative symptoms, physical state, comorbidities, and cognitive functioning”.
5. Difficulty is mentioned in the following sentence. The reason for this difficulty should be explained with the knowledge of the literature.
“It is evident that it is difficult to get closer to the meaning of happiness using measures and quantifiable instruments; because narratives are analyzed from a construtivist and phenomenological perspective, we lose relevant information that could help us reach our goal of understanding the meaning of happiness”.
5. It is not clear enough why this study is needed.
Methods
6. The characteristics of the place (cafeteria) where the interview was held? How was it standardized? eg. noise?
7. Expertise and experience of the interviewer?
8. Why was the focus group chosen? It should be explained
9. I think that the use of N-Vivo in analyzes adds strength to the study.
Results
10. The statements of the participants about the interviews should be given more space.
11. Figure 1 has too many spaces and the text is small. The figure should be revised.
Discussion
12. Limitations of the study should be added.
13. Suggestions that will guide future studies should be presented.
14. Only 3 references consist of studies done in the last 5 years. Discussion should be developed in the light of current literature.
15. Reference section should be revised. Underlined areas should be corrected.
Author Response

(The authors gave the same response as above.)

Round 2
Reviewer 2 Report
The purpose of the study can be better explained. Why was this study needed?
Author Response
1. The purpose of the study can be better explained. Why was this study needed?
"Happiness is important in achieving recovery in severe mental disorder [7], therefore, this study aims to approach the meaning of happiness for these people in order to improve their health, as their responses can guide nursing interventions."